# Lifestyle, Physical Activity and Dietary Habits of Medical Students of Wroclaw Medical University during the COVID-19 Pandemic

**DOI:** 10.3390/ijerph19127507

**Published:** 2022-06-19

**Authors:** Aureliusz Andrzej Kosendiak, Michał Piotr Wysocki, Paweł Piotr Krysiński

**Affiliations:** 1Department of Physical Education and Sport, Wroclaw Medical University, Wojciecha z Brudzewa 12, 51-601 Wrocław, Poland; aureliusz.kosendiak@umw.edu.pl; 2Wroclaw Medical University, wyb. Ludwika Pasteura 1, 50-367 Wrocław, Poland; pawel.krysinski@student.umw.edu.pl

**Keywords:** dietary habits, physical activity, medical students, coronavirus, nutritional knowledge, COVID-19, sleep, computer, lockdown

## Abstract

The new disease COVID-19, induced by SARS-CoV-2, causes acute respiratory infection. Many countries, including Poland, began to set a variety of different restrictions to reduce the spread of the virus. Most students had problems with online lessons. The study was conducted among second year medicine students of the Medical University of Wroclaw, and after the entire process of verification 200 respondents were accepted. The research consisted of completing the same anonymous online questionnaires twice in March and October 2020. This finally allowed for a critical assessment of the impact of the pandemic and its restrictions on the students’ daily lives. During the online classes, low levels of physical activity persisted (*p* = 0.718), whereas time spent sitting increased (*p* < 0.001). Despite positive changes in declared snacking (*p* = 0.061), we observed significant drops in the index of healthy diet (*p* = 0.001) and nutritional knowledge (*p* < 0.001) as well as an increase in the consumption of fast-food (*p* < 0.001) and energy drinks (*p* = 0.019). Reduced nutritional knowledge can cause a decrease in attention to healthy food preparation and much more frequent consumption of fast-food.

## 1. Introduction

The new disease COVID-19, induced by SARS-CoV-2, causes acute respiratory infection. COVID-19 was first reported in the city of Wuhan, China, began to spread rapidly around the world and, in March 2020, the WHO declared the coronavirus global outbreak a pandemic [1,2,3]. By 13 April 2022, according to data from Johns Hopkins University, there were 500,994,183 cases and 6,186,945 deaths from COVID-19. Poland, with 5,981,486 cases and 115,736 deaths, ranks 19th worst in the world [4].

In March 2020, an epidemic was declared in Poland. Restrictions were gradually tightened and at the peak they included closed shopping malls and sports facilities, highly limited service industry and the obligation of wearing masks in public. A national lockdown was announced—a ban on mobility, beyond basic living, health and work purposes. In addition, schools and universities in Poland were closed and distance learning was introduced [5,6]. The enforcement of such restrictions reduced the number of infections and flattened the infection rate curve so that the health service could still work effectively [7]. On April 20, along with the improving sanitary situation in Poland, the government introduced a gradual lifting of restrictions. Nevertheless, in the new academic year, in October, only medical universities, including ours in Wrocław, returned to live teaching. Unfortunately, at the end of the same month, the second wave of COVID-19 came and the government began to restrict movement again [5,6].

At that time, people’s professional work, lifestyle and ways of spending free time changed. Most employees switched to a remote form of work, which is conducive to reducing their daily activity. Moreover, during the lockdown people spent more time at home and modified their lifestyle by increasing the amount of time devoted to sitting [8]. Apart from the fact that on-line classes without direct contact were introduced at universities, physical education classes were also carried out remotely [9]. The study results of other authors showed that the social isolation caused by the pandemic reduced the level of physical activity, which increased the risk of other anti-health behaviours and could have contributed to deterioration of health as measured in the studied people [8,10].

Some of the major changes, caused by the pandemic, relate to eating habits [8,9,10,11,12]. People spend more time at home due to restrictions. Gathered data shows that Polish respondents reported eating more during the quarantine, and the majority of them admitted to snacking more frequently [11]. According to the study of Olfert et al. among American students during COVID-19, the pandemic not only increased the number of meals and snacks consumed per day, but also changed the size and frequency of meals, mostly because of boredom and stress [10]. On the other hand, the data from a university in Serbia presented by Sekulic et al. show that during the pandemic there was an increased consumption of fruits, vegetables and nuts, while eating of fast food decreased [8]. Compared to older adults, younger adults experienced higher consumption of unhealthy snacks and sugar-sweetened beverages, which the authors explain by possible lower exposure to stress among the older group of respondents. Dietary shifts to less healthy foods and drinks may influence metabolic health if sustained long-term [12,13].

The pandemic situation has a huge impact on mental health. This is due to the risk associated with the disease and the introduced restrictions. Additionally, frustration, boredom and financial losses were strong stressors [14,15]. Many people experienced severe mental stress as well as sleeping disorders [9,15,16]. The highest levels of depression and anxiety occurred in the early stages of lockdown but declined fairly rapidly, possibly because individuals adapted to the circumstances [17]. Young people and health workers were more likely to suffer from stress [18]. It was noted that students also experienced severe stressors and reported stress-related drinking, especially those reporting a low level of meaning in life [19]. There is currently a small number of studies examining thes3 changes in college-attending young adults. Our research assessed the level of physical activity, eating habits, addictions and lifestyle of medical students from the Medical University of Wrocław in March 2020 and 6 months later. The aim of the study was to compare these periods and make a critical assessment of the impact of the pandemic and its restrictions on the daily lives of students. It was hypothesised that all the parameters would deteriorate. Understanding how the situation influenced our respondents will facilitate adjusting classes, including physical education, to remove possible negative effects and to improve nutritional knowledge. Moreover, the study aims to diagnose problems that may arise during subsequent waves of infection or pandemic.

## 2. Materials and Methods

### 2.1. Study Design and Participants

The study was conducted among second year students at Wroclaw Medical University, taking part in compulsory physical education classes. The inclusion criteria are given in Figure 1. The participation in the study was voluntary. The age of the students was between 19 and 21 years old. All respondents were informed of their anonymity and their answers were used only in the study. The attendees gave their informed consent to participate in the study. It consisted of the same respondents filling in anonymous online questionnaires twice. Study participants had 2 weeks to complete them both. [20]. These questionnaires are research tools often used in publications and for the purposes of the study they have been modified into a remote version distributed using Google documents. The first took place in March 2020 and the second in October 2020. Exclusion and inclusion criteria are shown in the results section. In Phase 1, which lasted forv1 month, 310 people qualified for the study. Two questionnaires were used for the study: KomPAN and IPAQ. Additional series of questions were “metrics” and “COVID questions”. All responses were verified by the authors. Participants who had unreliable answers or did not answer the questions fully enough were removed. In Phase 2, which lasted for 2 months, 225 participants qualified after the first verification process. In Phase 3, additional exclusion criteria were introduced. As a result, 200 respondents were accepted after the entire verification process. The process of selecting participants is essential to a fairly conducted study. The research group consisted only of students at Wroclaw Medical University. Any failure to answer all questions in the questionnaire and chronic diseases in the respondents were exclusion criteria, because these results would be unreliable for this study [21]. The diagram below shows the entire process of preparation and selection of the study participants (Figure 1).

### 2.2. Level of Physical Activity

The Short Polish version of the International Physical Activity Questionnaire (IPAQ) can be used to measure and compare the physical activity of different populations. We used the short form to assess the level of physical activity among students. As the study shows, completion of the questionnaire by respondents may exaggerate the results, therefore all participants of the survey were trained by our research team [22]. The respondents answered the same questions in March and October 2020. Then we compared the results with each other. The results of the study were presented in energy expenditure expressed in MET units, i.e., the equivalent of resting metabolism. This is equal to the consumption of 3.5 mL of oxygen per kilogram of body weight per minute. In resting metabolism, MET level = 1. The questionnaire divided the respondents according to the following values: MET < 3.3 is low intensity; MET (>3.3 and <8) is moderate intensity; MET > 8 is high intensity [23].

### 2.3. Dietary Habits, Lifestyle, Diet Quality and Nutrition Knowledge

The research used the KomPAN questionnaire developed by the Committee of Human Nutrition Science Polish Academy of Sciences [24]. The questionnaire was created in order to learn about the eating habits and beliefs of Poles and used by us to determine the dietary habits, diet quality, lifestyle and nutrition knowledge of students.

Each of the respondents, on their own, filled in the questionnaire in the form of an online survey, consisting of 111 questions divided into 4 main topics: dietary habits (11 questions), frequency of food consumption (33 questions), nutrition beliefs (25 questions), lifestyle and personal data. Then, in accordance with the Kompan analysis instructions, 4 parameters were estimated: “Pro-Healthy Diet Index”, “Non-Healthy Diet Index”, physical activity and nutrition knowledge. Food frequency consumption was calculated in 6 categories (from “never”—(1) to “few times a day”—(6)). This was used to assess eating habits during the study. The frequency of food consumption was also converted into “times per day” (from “never”—(0) to “few times a day—(2)). Diet scores were calculated based on daily food consumption.

Pro-Healthy Diet Index-10 (pHDI-10) was calculated as the sum of the frequency of eating healthy food groups, included in 10 questions (no.: 23, 25, 31–33, 37, 38, 40, 42–43; the total score range: 0–20 points). Non-Healthy Diet Index-14 (nHDI-14) was calculated as the sum of the frequency of eating unhealthy food groups, included in 14 questions (no.: 22, 24, 26–29, 34–36, 44, 46, 51–52, 54; the total score range: 0–28 points).

Then these were summed up and classified according to the scale:pHDI—10: low = 0–6.66; moderate = 6.67–13.33; high = 13.34–20
nHDI—14: low = 0–9.33; moderate = 9.34–18.66; high = 18.67–28

Nutritional knowledge was estimated on the basis of 25 questions from Part C of the questionnaire. For each correct answer, the participant received 1 point, and at the end the result this was summed up and classified: insufficient = 0–8; sufficient = 9–16; good = 17–25. Another test to measure physical activity was used to check the lifestyle of students. This was estimated on the basis of 2 questions about activity in school/work and free time. Then the responses were combined into a single criterion as in Table 1.

### 2.4. Own Questions Questionnaires

Questions in the “metric” section contained basic information about the participants: age, gender, body height, weight and residence depending on the size of the city. Additionally, Body Mass Index was split. The results were compared over a 6 month period and used to divide the participants in the data analysis process. The last part of our questions was about the COVID-19 pandemic. BMI was calculated with the use of the data declared by the individual participants. It was not measured by the bio-impedancy method due to COVID limitations. This group of questions examined the impact of the pandemic on students in March 2020 and October 2020, and the difference in its influence as it progressed. The respondents answered questions about their status, i.e., restrictions imposed on them, well-being and real impact on themselves. The reliability of metrics and COVID-19 associated questions was calculated using Cronbach’s-Alpha test.

### 2.5. Statistical Analysis of Data

The data of were analysed with Microsoft Excel and Statistica 13.3 (Wroclaw Medical University’s license, Statsoft Polska Sp. z o.o., Wroclaw, Poland) package. The values of the continuous parameters were presented as the mean value and standard deviation (SD) and the categorical values were shown as the number and percentage. The values of the parameters included in questions about diet quality were changed to the values “times/day”. Body Mass Index was calculated according to WHO guidelines, but for the purpose of calculations, all of types of obesity constituted one group. The participants of our study are in the age range 19–21 and, in order to standardize the parameter, we calculated it in the same way as for adults, and some sources claim that BMI is calculated in this way already above 18 years of age [25]. Data between March and October were compared and analysed using the T-test for assessing the statistically significant differences. The χ^2^ (chi-square) test of independence was performed to assess statistically significant differences between expected and observed values contained in a contingency table. It was also used to check the dependence of two measured values over a 6 month period. After data analysis, the variances of the individual samples were compared using the F-test. The significance level of data analysis is 0.05. When the *p*-value was less than 0.05, the relationship between the variables was considered as statistically significant. The reliability of metrics and COVID-19 associated questions was calculated using Cronbach’s-Alpha.

## 3. Results

### 3.1. Anthropometric Data and Place of Residence

Comparing the period at the beginning of the pandemic and after six months, we noted changes close to statistical significance in the body weight measurements (*p* = 0.068) and the BMI values (*p* = 0.180). The average body weight of the study participants was 63.9 ± 11.5 kg for the first measurement and 64.3 ± 11.5 kg for the second. We recorded an increase among 37% of respondents. At the time of the first measurement, 31 (15.5%) respondents had abnormal BMI—including 16 (8%) people who were underweight. Although the overall number of people with abnormal BMI practically remained the same (30–15%), we found a marked decrease of 18.75% in the percentage of people who were underweight. Interesting results were recorded in place of residence (*p* = 0.001). Between the surveys, the respondents headed to places of residence with a larger population, and the biggest increase occurred in the number of people living in cities in the area of 500,000 population—from 38 to 52. Characteristics of the study group are presented in Table 2.

### 3.2. Measurements of Physical Activity

When it comes to physical activity, we did not find statistically significant differences in its level, measured in KomPAN (*p* = 0.718) and IPAQ (*p* = 0.181). However, in both forms we showed the same trend—a decrease in the number of people with moderate physical activity (KomPAN: 78 to 72 and IPAQ: 40 to 17) and an increase with low or high physical activity. On the other hand, there is a statistically significant change in the time spent on walking (*p* < 0.001) and sitting (*p* < 0.001). The time devoted to these activities increased by 90.93% and 60.43% accordingly (Table 3).

### 3.3. Healthy and Unhealthy Diet Indexes, Nutritional Knowledge

During the COVID-19 pandemic, we recorded statistically significant drops in the average pro-health diet index and the level of nutritional knowledge, of 8.97% (*p*= 0.001) and 15.3% (*p* < 0.001), respectively. The number of people with good nutritional knowledge significantly decreased (73 to 35) and the number of people with insufficient knowledge increased (9 to 42). Significant differences in the pro-health diet index were caused by a drop in the number of people with a moderate index (44 to 22) and an increase of those with a low index (156 to 178). It is worth noting that among the surveyed second year students we did not notice any person with a large pro-health index in any of the periods. However, contrary to the above parameters, we did not notice any significant statistical changes in the unhealthy diet index (*p*= 0.649) (Table 4).

### 3.4. Relationship between Diet Indexes and Nutritional Knowledge

Additionally, we compared the level of nutritional knowledge with the pro-health diet and unhealthy diet indexes. In March we found a statistically significant relationship between nutritional knowledge and the pro-health diet index (*p* = 0.02). People with sufficient knowledge had a higher average pro-health index by 14.68%. The unhealthy diet index was unchanged regardless of nutritional knowledge. In October we did not notice the statistical relationship between nutritional knowledge and the pro-health diet index (*p* = 0.594) and unhealthy diet index (*p* = 0.279). However, nutritional knowledge correlates with the average pro-health diet index (insufficient—1.07; sufficient—1.11; good—1.14).

### 3.5. Frequency of Food Consumption

Comparing the period at the beginning of the pandemic and after six months, we did not find statistically significant changes in the declared number of meals during the day (*p* = 0.716) and general regularity (*p* = 0.435). We observed a low percentage of people consuming less than three meals a day in both the first (7%) and the second (10%) studies. On the other hand, the number of people who eat all meals regularly, amounting to 40 and 35 people in the first and second studies respectively, seems alarmingly low. A nearly significant difference occurred in the amount of snacking (*p* = 0.061). It is worth noting that we noted a clear decrease from 50% to nearly 40% in the percentage of people eating between meals once or several times a day (Table 5).

### 3.6. Fast Food and Energy Drinks Consumption

Significant changes were observed in the declared consumption of fast-food (*p* < 0.001) and energy drinks (*p* = 0.019). Their average consumption increased accordingly by 30% and 14%. When it comes to fast-food, the percentage of people consuming them 1–3 times per month has significantly decreased (64.5% is 35.5%), and increased by several times a week (6% to 27%) and once a week (16.5% to 27%). On the other hand, referring to energy drinks, the number of people never drinking them decreased (68.5% to 55%), and increased by 1–3 times per month (22.5% to 29%) and once per week (6% to 13.5%) [Table 5].

### 3.7. Alcohol Consumption

Interesting differences occurred in the frequency of alcohol consumption (*p* < 0.001)—it decreased by 20%. This result was influenced by the increase in the number of students declaring no consumption (44 to 107), and the decrease in those who declared consumption 1–3 times per month (101 to 60), once per week (39 to 22) and a few times per week (15 to 9). The opposite is the case with smoking tobacco products (*p* = 0.003)—addiction was declared by over twice as many people (11 to 26). Interestingly, 9 out of the original 11 smokers managed to quit and therefore we recorded as many as 24 people who fell into this category. The results are presented in Table 5.

### 3.8. Pandemic Restrictions, TV and Computer, Quality of Sleep

During the pandemic, the restrictions imposed on our students changed (*p* = 0.002). The number of people under no restrictions rose significantly (39 to 70) and the number of people under only government restrictions (155 to 124) decreased. On the other hand, the percentage of people subjected to quarantine or epidemiological surveillance remained unchanged and amounted to 3%. Very significant differences occurred in the responses to the impact of the pandemic (*p* < 0.001). There was a marked increase in the number of people who rated a negative impact on their lives (79 to 117), and a drop in the number of people who were positively affected by the pandemic (16 to 13), were not influenced (27 to 18) or were unable to answer this question (78 to 52). No significant changes were recorded in the responses to the questions relating to the time allocated to sleep (*p* = 0.922) and activity in front of a computer, TV or other electronic devices (*p* = 0.801). In the case of sleep, it is important that the vast majority of students reported 7 or more hours of sleep. In both studies, this figure is 74.5% and 75.5%. However, the amount of time spent in front of electronic devices is disturbing. Nearly 60% of respondents spent between 4 and 8 h a day in front of the computer in both phases. The results are presented in Table 6.

## 4. Discussion

### 4.1. Weight, BMI, Place of Residence

The present study investigated the impact of the pandemic and restrictions on physical activity, eating habits, addictions and lifestyle of medicine students. According to the Government Population Council, in 2020 the trend of population growth in large agglomerations and villages continued, while the population of medium-sized cities and towns decreased [26]. Our statistically significant results show that, although respondents were targeting large agglomerations, when it comes to villages and medium-sized cities we have different situations. Living in the countryside was declared by fewer people than at the beginning of the pandemic, while in cities of 100–500k the number was the same, and in cities of 50–100k it increased. This shows that during the pandemic, medical students were heading to larger and medium-sized urban centres, which is possibly caused by the need for education in health care units that are located there.

As for the measurement of weight and BMI, in both cases we obtained statistically insignificant results, but the measurement of weight was very close. We recorded an average increase in weight by 0.4 kg, while the increase was declared by more than one third of the respondents, and a decrease in body weight by 25% of people. Data from the Polish population, presented by Sidor et al., show that the increase in body weight occurred in 30% and the decrease in 18% of the Poles [11]. On the other hand, the work of Sumalla-Cano et al. among university students in Spain shows an increase in weight in 35.7% and a decrease in 36.3% of respondents [3], and data from the Spanish population presented by Sanchez et al. report an increase in weight in 52.7% of the population [27]. This shows that the percentage of our respondents who noticed a change in weight (62%) was higher than in the Polish population by more than 25%, while the difference between people with weight loss and increase was similar. Comparing our data to those from a Spanish university, we observe a significantly lower percentage of people with weight loss (by nearly 50%), a similar percentage of people with an increase in body weight and a lower percentage of people with a change in weight overall (by nearly 12%). In relation to the Spanish population, there is a lower percentage of people with an increase in weight (by 14.2%).

When it comes to BMI, its incorrect level was noticed in approximately 15% of the respondents in both the first and the second survey. This is an excellent result and lower in comparison with the studies of the Polish (59%), Spanish (65%), Chilean (71%) and Dutch (73%) populations. A positive signal was also the decrease in the percentage of underweight students by almost 20% (6.5%), which made it better compared to the Polish population (7.9%), but unfortunately significantly worse compared to the studies in Spain (2.9%) [27,28,29].

### 4.2. Nutritional Knowledge, Dietary Habits, Meals, Fast Food, Energy Drinks

The COVID-19 pandemic impacted the dietary habits of people. The data from the Polish population shows that 43.5% of the respondents reported eating more during the quarantine, and the majority admitted to snacking more frequently. Moreover, almost one third of those surveyed did not consume fresh vegetables and fruits on a daily basis, but the same percentage ate sweets at least once a day [11]. Similar changes were observed in a study among US adults, in which 36% of adults reported sometimes consuming more unhealthy snacks or desserts, while 16% did so often or always. Compared to older adults, younger adults experienced higher consumption of unhealthy snacks and sugar-sweetened beverages, which the authors explain by possible lower exposure to stress among the older group of respondents [12].

According to the study of Olfert et al. among US students during COVID-19, the pandemic not only increased the number of meals and snacks consumed per day, but also changed the size and frequency of meals. The study shows that 42.9% of students reported eating more, and mostly this was because of boredom and stress. Additionally, more than a quarter of respondents indicated more frequent cooking, and most of them described this as healthy. However, the authors recorded an increased consumption of healthy food (fruits, vegetables and legumes) and unhealthy food (red and processed meats) [10]. The data from a university in Serbia presented by Sekulic et al. show that, during the pandemic, there was an increased consumption of fruits, vegetables and nuts, while the use of fast food decreased. Moreover, the authors point to the increase in the number of meals in general and the use of dietary supplements [8]. On the other hand, the work of Gadi et al. among students in the UK reports diet changes in more than half of the participants, with most experiencing a decrease in quality [9].

During the COVID-19 pandemic, we also recorded the impact on dietary habits. Unfortunately, the trend is similar to works from the Polish population, US adults and university students in the UK [9,11,12]. Although there were no significant changes in the unhealthy diet index and the frequency of meals, the nutritional knowledge of medicine students deteriorated. The number of people with good nutritional knowledge decreased from 73 to 35, and with insufficient knowledge increased from 9 to 42. The pro-health diet index decreased among the respondents. The number of people with a moderate index decreased from 44 to 22, and with a low index increased from 156 to 178. Among the group of 200 respondents, these are quite significant differences, taking into account the 6-month period. Additionally, there was a relationship between nutritional knowledge and the pro-health index in the first study. Sufficient knowledge correlated with a higher pro-health index. We also observed a change close to statistical significant in declared snacking. In contrast to data from the Polish population, declared snacking decreased by an average of 12%. Moreover, we recorded a clear decrease from 50% to nearly 40% in the percentage of people eating between meals once or several times a day. Unfortunately, contrary to data from a university in Serbia, statistically significant increases during the COVID-19 pandemic were observed in the declared consumption of fast-food (*p* < 0.001). Additionally, the same situation was recorded with energy drinks (*p* = 0.019) [Table 5]. Their average consumption increased accordingly by 30% and 14%.

To sum up, despite positive changes in declared snacking, we observed significant drops in the index of healthy diet and nutritional knowledge, and an increase in the consumption of fast-food and energy drinks. Less nutritional knowledge can be caused by decreased attention to healthy food preparation and more frequent consumption of fast-food.

### 4.3. Physical Activity

The role of physical activity in health is undeniable. There are many studies confirming that the level of physical activity affects mood and well-being [30,31,32,33]. In March 2020, the World Health Organization declared a pandemic [1,2,3]. At that time, people’s behaviour related to professional work, lifestyle and forms of spending free time changed. Most of working people switched to a remote form of work, which is conducive to reducing their daily activity. During the lockdown people spent more time at home and modified their lifestyle by increasing the amount of time devoted to sitting [8]. Moreover, on-line classes without direct contact were introduced at universities and physical education classes were also carried out remotely. Overall decreased level of physical activity in various social groups was observed, including among students. This fact was partly due to the introduced restrictions, bans and closings of places where various forms of physical activity could be carried out. During the pandemic, swimming pools, gyms and fitness centres were closed. Depending on the intensity of the disease, various types of defence were introduced, as well as limits on people who could participate in sports events [10].

In the authors’ own research, the level of physical activity was assessed using the IPAQ questionnaire as well as the selected part of the KomPAN questionnaire. In contrast to the above studies, in both questionnaires we did not find statistically significant differences in its level. Despite this fact, we showed the same trend, a decrease in the number of people with moderate physical activity and an increase in low or high physical activity. Moreover, special attention should be paid to the fact that medical students do not perform their physical activity at the appropriate recommended level.

It is worth noting that an alarmingly high percentage of respondents declared a low level of activity. In the IPAQ questionnaire at the beginning of the outoutbreakthis was 68% of the surveyed students, and after 6 months it was 79%. When it comes to the KomPAN questionnaire, it was about 60% in both periods. The percentage of respondents who had a high level of physical activity was also disturbing. In the IPAQ and the KomPAN, the maximum was 12.5% and 3% accordingly.

The conducted research also assessed the number of minutes per day spent on walking and sitting. Most of the studies confirmed the fact that the amount of time spent walking during the day decreased due to the epidemic [33]. Interestingly, in my own research, an inverse relationship was observed. At the beginning of the epidemic, i.e., in March 2020, students spent an average of only 25.3 min a day walking. After six months of the epidemic, the study was repeated on the same group, and it turned out that the same students rated their walking level at 48.3 min a day. The change in the amount of time spent walking was highly statistically significant.

This state of affairs may have been due to the fact that, in March 2020, at the beginning of the epidemic, numerous restrictions were introduced and the classes at the Medical University of Wroclaw were conducted only online. Students did not move around going to individual classes, which took place through special e-learning platforms in the comfort of their own home or lodgings. In October 2020, classes in a hybrid form were introduced and students returned to the university, which could be associated with greater activity in the form of, for example, moving from class to class.

The studies also looked at the amount of time spent sitting. Measurement at two time intervals at the beginning and after six months of the epidemic showed a statistically significant relationship. It was observed among medical students that the amount of time spent in a sitting position increased by as much as 60% after 6 months of the epidemic in relation to the time of its outbreak. Perhaps such a state of affairs resulted from the situation of the beginning of the academic year, where in this initial stage students devoted a lot of time to study.

### 4.4. Alcohol and Cigarettes

The consequence of smoking and drinking alcohol can be numerous health ailments or chronic diseases related to excessive consumption, as with the use of all kinds of stimulants. Drinking alcohol can be one way of dealing with stress, regulating tension, and avoiding confrontation with problems [11,34]. During the pandemic, a number of studies were conducted to assess alcohol consumption and cigarette smoking in various social groups [13,35,36,37]. Interestingly, most studies showed an increase in alcohol consumption during the epidemic, which may confirm the fact of fear and anxiety. Outbreaks related to the duration of the epidemic may be factors that led to an increase in alcohol consumption [13,36,38]. In our study, we obtained different results. It was observed that alcohol consumption decreased by almost 20% after 6 months of the epidemic. The observed relationship was statistically significant. The results may be dependent on the fact that students started the new academic year (October 2020) and consumed less alcohol compared to the time (March 2020) when there was complete lockdown isolation. However, it should be emphasized that alcohol consumption was assessed using the KomPAN questionnaire and the provided answers were the declarations of the research participants, therefore the obtained data should be interpreted with caution.

Smoking cigarettes is perhaps the most common consumption of a drug. More and more people in the world use nicotine, not only in the form of smoking, but also cigars and the increasingly fashionable e-cigarettes. According to the World Health Organisation, tobacco kills more than eight million people each year, including both smokers and non-smokers who are exposed to passive smoking. In our study, an increased number of people smoking cigarettes was observed, which was also confirmed by the results of studies by other authors [36,39,40,41]. After 6 months of the pandemic, an over two-fold increase in the number of smokers was observed, despite the fact that a part of the research group declared having quit smoking.

### 4.5. Use of Computers, Sleep Quality and Impact of Pandemic

All changes related to the pandemic had a significant impact on mental health [18], sleeping [9,16], mode of work and education [7]. Our students were also restricted and transitioned into e-learning. In our study, we obtained statistically significant changes in the restrictions imposed on the respondents. Part 1 of the study was conducted at the end of March 2020, so nearly 2 weeks after “going online”. October 2020, on the other hand, is the period before the second wave of the pandemic in Poland, as well as the normal return of medical students to live education. Therefore, it is not surprising that there was an increase in the number of unrestricted respondents and a stable level of students in quarantine and isolation [Table 6].

When it comes to the impact of the pandemic, a study among students at a UK university shows that 84.2% of those surveyed reported worrying too much about different things, 61.9% could not stop or control their worrying, 72.1% felt unable to cope with things they had to do at least sometimes and 71.2% experienced trouble relaxing on several days or more [9]. The study from China highlighted the overall prevalence of generalized anxiety disorders, depressive symptoms, and sleep quality of the public on the level of 35.1%, 20.1%, and 18.2%, respectively and a significantly higher prevalence of generalized anxiety disorders and depressive symptoms among younger than older people [18]. Unfortunately, a similar tendency and statistically significant changes in the perception of the pandemic were also noted among our students. There was an increase of nearly 50% in people who are negatively affected by the pandemic, there was a persistently low percentage of people who are positively affected by the pandemic, and there was a marked decline in the number of people who are not affected or cannot respond to the question. The work of Brooks et al. reports negative psychological effects of the COVID-19 pandemic including post-traumatic stress symptoms, confusion, and anger caused by stressors, such as longer quarantine duration, infection fears, frustration, boredom, inadequate supplies, inadequate information, financial loss, and stigma [14]. Moreover, when quarantine is necessary, the authors point out the important role of thoroughly informing the public about the reasons for the restrictions applied, their procedures, and making these tolerable [42].

In our research we also evaluated the amount of time spent in front of the computer or television. As a result of numerous restrictions, bans and the introduction of remote classes, it became natural that students spent more time in front of the computer or television. In the work of Jayanti Mishra et al., who also conducted research among medical students in a similar period, more than 65% spent more than 8 h in front of a computer per day. The study also showed that more than 50% of those who spent this amount of time in front of a computer had a poor quality of sleep, and in the group of people who spent less than 8 h a day in front of a computer, this percentage was just over 33% [43]. Another study, conducted in 66 secondary schools in Hungary among students, also showed an increase in time spent in front of the computer and television as a consequence of the introduction of distance learning [44]. An increase in sedentary lifestyle, especially in front of the monitor, was also observed by research in Italian, Spanish, German and Australian students, while physical activity significantly decreased in these groups [45,46,47,48,49,50]. Similar correlations were found in Chwalczynska’s study, where an increase in the number of hours spent in front of the monitor during lockdown resulted in an increase in the amount of body fat in the students of the Medical University of Wrocław and the University of Physical Education. [51] This fact was confirmed by the results showing a more than twofold increase in the amount of time spent in front of a computer or TV, as shown in Table 6.

However, the results of our work did not show a statistically significant increase in time spent in front of a computer among medical students regardless of the study period, i.e., March 2020 and October 2020, which may be due to the fact that, already from the beginning of the epidemic as well as after 6 months of its duration, medical students were spending a lot of time on it every day. Nearly 60% of users spent between 4 and 8 h a day and every tenth student spent more than 8 h a day in front of the computer in both periods. It should also be noted that increased time spent in front of a computer can lead to reduced physical activity, change in eating habits and affect the quality of sleep, which is also important for health.

The study conducted during the COVID-19 pandemic also assessed the quality and quantity of sleep in different social groups. A number of authors observed a deterioration in the quality of sleep as well as a decrease in its length. Adequate sleep is essential for the proper functioning of the entire organism. As shown earlier and confirmed by the results of the research, the situation of the announcement of the pandemic and the lockdown which was its consequence led to a change in the lifestyle of individuals, and increased stress which also affected the quality of sleep. Medical students were a frequently studied group during the COVID-19 epidemic in terms of sleep quality [52,53,54,55].

In our study, we only assessed the duration of sleep in part of the KomPAN questionnaire. It is important that the vast majority of students reported 7 or more hours of sleep in both periods, which was 74.5% and 75.5%, accordingly. However, almost 25% of students reported less than 6 h of sleep per night, regardless of the duration of the pandemic. This change was statistically insignificant; however, it is worth noting that it was an insufficient amount for people at this age [56].

### 4.6. Limitations of the Study

Presented research highlighted changes in key COVID-19-related outcomes. However, our results had only an informative character. The study group consisted of medicine students, who are only a narrow part of the Polish population. It should be taken into consideration that the results of the questionnaire research were only based on the declarations of the participants, therefore they should be interpreted carefully. Moreover, the individuals selected to the study group had no chronic metabolic or psychological diseases. On the other hand, the research conducted in this group is pretty hard to compare with other results because of the poor information in the literature about similar studies focused on students, especially those from medical universities. Taking into consideration all the above, the COVID-19 impact on the studied behavioural factors cannot be extrapolated to the adult human population. It seems advisable to broaden the research to include other social groups, as well as to increase the number of respondents.

## 5. Conclusions

The COVID-19 pandemic and its restrictions had a serious impact on the daily lives of the students. There was a huge increase in the number of people negatively affected by this period. During the online classes, low levels of physical activity persisted (*p* = 0.718) and time spent sitting increased (*p* < 0.001). Despite the positive changes in declared snacking (*p* = 0.061), we observed significant drops in the index of a healthy diet (*p* = 0.001) or nutritional knowledge (*p* < 0.001), and on the other hand an increase in the consumption of fast-food (*p* < 0.001) and energy drinks (*p* = 0.019). A reduced nutritional knowledge level can be caused by a decrease in attention to healthy food preparation and much more frequent consumption of fast-food.

## Figures and Tables

**Figure 1 ijerph-19-07507-f001:**
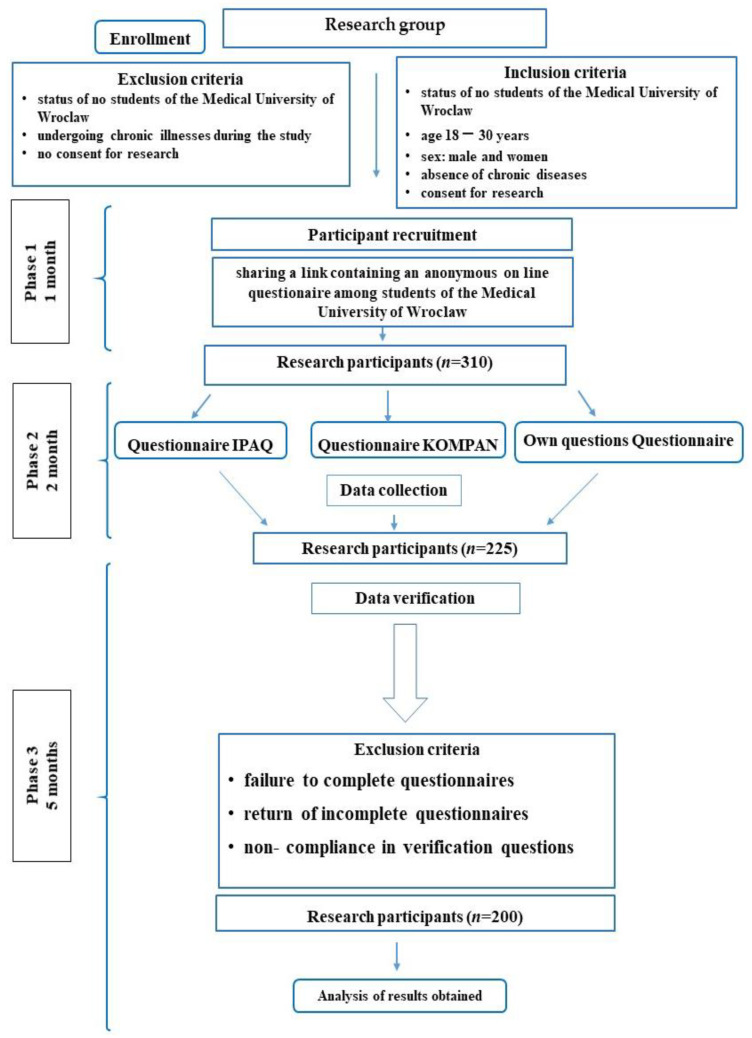
Selection process of the study.

**Table 1 ijerph-19-07507-t001:** Assessment of daily activity.

Activity in School/Work		Activity in Free Time	
	Low	Moderate	High
Low	**Low**	**Low**	**Medium**
Moderate	**Low**	**Medium**	**Medium**
High	**Medium**	**Medium**	**High**

**Table 2 ijerph-19-07507-t002:** Characteristics of the study group.

	March 2020 (*n* = 200)	October 2020 (*n* = 200)	*p*-Value
Sex			N.A.
Male Female	65 (32.5%) 135 (67.5%)	65 (32.5%) 135 (67.5%)	
Age (years) (Mean ± SD)	19.5 ± 0.6	19.9 ± 0.6	N.A.
Height (centimetres) (Mean ± SD)	171.9 ± 9	171.9 ± 9	N.A.
Weight (kilograms) (Mean ± SD)	63.9 ± 11.5	64.3 ± 11.5	0.068
Body Mass Index (Mean ± SD)			0.180
<18.5 Underweight 18.5–24.9 Normal	16 (8%) 169 (84.5%)	13 (6.5%) 170 (85%)	
Weight			
25.0–29.9	13 (6.5%)	14 (7%)	
Overweight			
>29.9 Obese	2 (1%)	3 (1.5%)	
Inhabitancy			0.001
City > 500,000	38 (19%)	52 (26%)	
City 500,000–100,000	25 (12.5%)	25 (12.5%)	
Town 100,000–50,000	24 (12%)	27 (13.5%)	
Town 50,000–20,000	34 (17%)	28 (14%)	
Town 20,000–5000	15 (7.5%)	10 (5%)	
Village	64 (32%)	58 (29%)	

**Table 3 ijerph-19-07507-t003:** Level of physical activity.

	March 2020 (*n* = 200)	October 2020 (*n* = 200)	*p*-Value
International Physical Activity Questionnaire (IPAQ)			0.181
Low	137 (68.5%)	158 (79%)	
Average	40 (20%)	17 (8.5%)	
High	23 (11.5%)	25 (12.5%)	
How much time did you average spend walking during the day? (minutes) (Mean ± SD)	25.3 ± 62.9	48.3 ± 45.5	<0.001
How much time did you average spend sitting during the day? (minutes) (Mean ± SD)	251.5 ± 507.8	403.4 ± 330.5	<0.001
Physical activity (from KomPAN)			0.718
Low	128 (64%)	122 (61%)	
Moderate	71 (35.5%)	72 (36%)	
High	1 (0.5%)	6 (3%)	

**Table 4 ijerph-19-07507-t004:** Diet quality and nutritional knowledge.

	March 2020 (*n* = 200)	October 2020 (*n* = 200)	*p*-Value
Pro-Healthy Diet Index (pHDI-10)			0.001
Low	156 (78%)	178 (89%)	
Average	44 (22%)	22 (11%)	
High	0	0	
Non-Healthy Diet Index (nHDI-14)			0.649
Low	197 (98.5%)	196 (98%)	
Average	3 (1.5%)	4 (2%)	
High	0 (0%)	0 (0%)	
Level of nutritional knowledge			<0.001
Insufficient	9 (4.5%)	42 (21%)	
Sufficient	118 (59%)	123 (61.5%)	
Good	73 (36.5%)	35 (17.5%)	

**Table 5 ijerph-19-07507-t005:** Eating habits, alcohol and cigarettes.

	March 2020 (*n* = 200)	October 2020 (*n* = 200)	*p*-Value
How many meals do you usually consume daily?			0.716
1	1 (0.5%)	0	
2	13 (6.5%)	20 (10%)	
3	77 (38.5%)	69 (34.5%)	
4	78 (39%)	81 (40.5%)	
5 or more	31 (15.5%)	30 (15%)	
Do you consume meals at a regular time?			0.435
No	56 (28%)	62 (31%)	
Yes, but only some of them	104 (52%)	103 (51.5%)	
Yes, all of them	40 (20%)	35 (17.5%)	
How often do you snack between the meals ?			0.061
Never	6 (3%)	8 (4%)	
1–3 times per month	7 (3.5%)	12 (6%)	
Once per week	20 (10%)	22 (11%)	
Few times per week	67 (33.5%)	75 (37.5%)	
Once per day	43 (21.5%)	37 (18.5%)	
Few times per day	57 (28.5%)	46 (23%)	
How often do you drink energy drinks ?			0.019
Never	137 (68.5%)	110 (55%)	
1–3 times per month	45 (22.5%)	58 (29%)	
Once per week	12 (6%)	27 (13.5%)	
Few times per week	5 (2.5%)	4 (2%)	
Once per day	0	1 (0.5%)	
Few times per day	1 (0.5%)	0	
How often do you eat fast food ?			<0.001
Never	25 (12.5%)	16 (8%)	
1–3 times per month	129 (64.5%)	71 (35.5%)	
Once per week	33 (16.5%)	54 (27%)	
Few times per week	12 (6%)	54 (27%)	
Once per day	1 (0.5%)	4 (2%)	
Few times per day	0	1 (0.5%)	
How often do you drink alcohol ?			<0.001
Never	44 (22%)	107 (53.5%)	
1–3 times per month	101 (50.5%)	60 (30%)	
Once per week	39 (19.5%)	22 (11%)	
Few times per week	15 (7.5%)	9 (4.5%)	
Once per day	0	1 (0.5%)	
Few times per day	1 (0.5%)	1 (0.5%)	
Smoking (No. and %)			0.004
No	189 (94.5%)	174 (87%)	
Yes	11 (5.5%)	26 (13%)	

**Table 6 ijerph-19-07507-t006:** Use of computers, sleep quality and impact of pandemic.

	Mar-20	Oct-20	*p*-Value
	(*n* = 200)	(*n* = 200)	
How many hours per day do you spend on average sleeping?			0.922
6 and less	51 (25.5%)	49 (24.5%)	
From 6 to 9	137 (68.5%)	140 (70%)	
9 and more	12 (6%)	11 (5.5%)	
How many hours per day do you spend on average watching TV or in front of a computer?			0.801
2 and less	14 (7%)	25 (12.5%)	
From 2 to 4	45 (22.5%)	31 (15.5%)	
From 4 to 6	59 (29.5%)	60 (30%)	
From 6 to 8	57 (28.5%)	57 (28.5%)	
From 8 to 10	17 (8.5%)	21 (10.5%)	
10 and more	8 (4%)	6 (3%)	
Has the pandemic situation affected you?			<0.001
Negatively	79 (39.5%)	117 (58.5%)	
Positively	16 (8%)	13 (6.5%)	
Unaffected	27 (13.5%)	18 (9%)	
No opinion	78 (39%)	52 (26%)	
Are you subject to any pandemic restrictions?			0.002
No restrictions	39 (19.5%)	70 (35%)	
Only government restrictions	155 (77.5%)	124 (62%)	
Quarantine	4 (2%)	6 (3%)	
Epidemiological surveillance	2 (1%)	0	

## Data Availability

The research results presented are part of a large ongoing study which has not yet been completed. If you are interested in specific data, please contact the corresponding author.

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
