# Peer review of "Lifestyle, Physical Activity and Dietary Habits of Medical Students of Wroclaw Medical University during the COVID-19 Pandemic"

_ijerph, 2022, doi:10.3390/ijerph19127507_

Round 1
Reviewer 1 Report
I think it is a very timely paper that examines the changes of college students in the pandemic situation.
However, there are some academic aspects that are not revealed. This needs to be corrected.
In the introduction part, there is a part where you describe your opinion as if it were true without academic evidence.
And it contains content that is acceptable in some areas but may not be applicable globally.
The content of the introduction is mainly about changes in the environment related to factors. The need for research and the awareness of the problem should be included in general.
Please present the expected academic value and meaning of this study.
The purpose of the study described at the end of the introduction is close to the research contents rather than purpose. Please make your point clear through this study.
Please describe your research questions.
Please add an additional description of why Wroclaw Medical Univ's study for students is meaningful.
Clarify the criteria for selecting the research participant in the research method.
Add an explanation of why the number of study participants has decreased.(310->200)And please add the text related to the effect (or no effect) that this decrease in the number of research participants can have on the results of the research based on previous studies
The time range for this study is March to October 2020. Please add an additional explanation of the Pandemic situation in the country or region to which the study participants belong
Author Response
we would like to thank you for precious and helpful comments, which undoubtly allowed us to improve our article. We carefully analyzed each of them and modified the text accordingly.

Reviewer 2 Report
Dear Authors of the manuscript Lifestyle, physical activity and dietary habits of medical students of Wroclaw Medical University during the Covid-19 pandemic.
I have read the report of your study with interest, and I believe it may deserve to be published on the IJERPH if you are willing to do the following major revisions:
An extensive linguistic editing of the text is absolutely required to improve its readibility.
The manuscript is unnecessarily long, in particular Introduction and much more the Discussion contain many repetitive sentences The Discussion must be shortened a lot.
Abstract:
report the main results with the related statistical estimates
do not write conclusions that reflect your thoughts, only the conclusions drawn from your data
Introduction
some sentence requires a different or further citation, please see the attached file
The introduction seems the result of many sentences that are not always well connected to each other. Eg. the penultimate paragraph that talks about online lessons is not well connected to the previous ones and I think it is useless as this theme is not central in your results. Moreover, the objective of the study is not well motivated. Overall I suggest to be a little more concise in the introduction and better motivate why you wanted to do this study.
Create a new Figure 1 with better resolution. Colors are not necessary, they actually create troubles in reading
Methods
Was the study approved by the ethics commette? if not, based on which law?
How were the students approached?
Where could the questionnaire be accessible?
How was the study advertised among students?
What information were shared with students about the questionnaire? there was an informed consent?
Discussion
please, be more concise an do an extensive English editing. Also check acronym and punctuation
Check the reference completness
Please, be aware the without a professional editing and a massive rationalization of contents (especially in the Discussion and a little in the Introduction) the manuscript is very difficult to read and the readers struggle to make sense of the content. So in this form it is not acceptable

Author Response

(The authors gave the same response as above.)

Reviewer 3 Report
Dear authors,
The paper entitled "Lifestyle, physical activity and dietary habits of medical students of Wroclaw Medical University during the Covid-19 pandemic”, has been revised. I will make some suggestions to improve the manuscript.
Include in the abstract the size of the universe and the study design. Adjust the abstract according to the observations detailed in the other sections.
I suggest strengthening the introduction. 1) Incorporate evidence from other populations about lifestyle changes resulting from the pandemic; many of the studies reviewed in the discussion may be helpful; 2) The objective is unclear. It is not clear whether they intend to describe, compare, or determine the impact of the pandemic.
Material and methods: 1) Clarify the study design; 2) Point out the size of the universe; 3) Describe how they maintained the anonymity of the participants and were able to contact them for the second measurement; 4) Report the loss of participants throughout the study; 5) Describe the psychometric properties of the scales used; 6) Detail the age range of the participants; 7) The criteria used to classify BMI are for a population aged 20 years or older, so the estimation should be made according to percentiles and not by direct BMI.
Results: 1) Table 3 needs to clarify that the time is presented in minutes; 2) It is suggested to apply post hoc tests where applicable.
Discussion: 1) Considering as a limitation that obtaining bodyweight by self-reporting may be a biased response; 2) Arguing how the loss of knowledge can occur since according to their results "The number of people with good nutritional knowledge decreased from 73 to 35..."; 3) It is not clear why they classify the study as cross-sectional ("line 586. .our results were only of a cross-sectional and informative character...) if they measured on two occasions the same variables in the same people (line 84 "...It consisted of filling in anonymous online questionnaires, which were performed among the same respondents...").
The conclusions assert variables that were not included in the study such as stress and coping with stress “Negative changes in dietary habits can be associated with the high stress placed on young people and in the long run, poor diet quality leads to weaker health outcomes. One way of coping with stress resulting from the pandemic condition was to use cigarettes more frequently.”
Author Response

(The authors gave the same response as above.)

Round 2
Reviewer 1 Report
The overall content has been modified appropriately. However, please consider expanding the academic meaning of the study a little more. I think it will be a better study if the contents of why this study is meaningful and what is the difference from other studies are further supplemented. For this, the in-depth review of previous studies is needed.
Reviewer 2 Report
Dear Authors, thank you for having Co sidered the suggestions provided.
Reviewer 3 Report
My comments have been addressed